# Association between the Perception of Behavior Change and Habitual Exercise during COVID-19: A Cross-Sectional Online Survey in Japan

**DOI:** 10.3390/ijerph20010356

**Published:** 2022-12-26

**Authors:** Daisaku Nishimoto, Shimpei Kodama, Ikuko Nishio, Hyuma Makizako

**Affiliations:** 1Department of Nursing, School of Health Sciences, Faculty of Medicine, Kagoshima University, Sakuragaoka 8-35-1, Kagoshima 890-8544, Japan; 2Department of Physical Therapy, School of Health Sciences, Faculty of Medicine, Kagoshima University, Sakuragaoka 8-35-1, Kagoshima 890-8544, Japan

**Keywords:** cross-sectional study, internet use, exercise, transtheoretical model, pandemics

## Abstract

In general, the perception of behavior change may be associated with habitual exercise. However, this association might not be well-understood due to the state of emergency of the COVID-19 pandemic. This study collected data from 1499 internet users aged 20–86 years living in Japan who participated in the online survey from 26 to 27 February 2021. Having a perception of behavior change was defined as preparation, action, and maintenance of the transtheoretical model. The habitual exercise was defined as 600 metabolic equivalent min/week or more based on the International Physical Activity Questionnaire. Multivariate logistic regression analysis was used to calculate the odds ratio of habitual exercise and a 95% confidence interval was estimated after adjusting for related factors. We found that perception of behavior change was positively associated with habitual exercise (adjusted odds ratio = 2.41, 95%CI = 1.89–3.08), and similar associations were found in states of emergency (2.69, 1.97–3.69) and non-emergency (2.01, 1.34–3.01). Moreover, women were negatively associated in all analyses with habitual exercise compared to men (0.63, 0.49–0.80; 0.65, 0.44–0.96; 0.62, and 0.45–0.84, respectively). Thus, the perception of behavior change may be involved in the implementation of habitual exercise, suggesting that women face difficulties in engaging in habitual exercise.

## 1. Introduction

The COVID-19 infection is reported to have high mortality due to the fact that it is a highly contagious disease with a strong negative association with comorbidities such as diabetes, cancer, hypertension, and coronary heart disease [1]. Prior to the COVID-19 outbreak, the impact of non-communicable diseases (NCDs) on deaths was found to be significant, with approximately 70% of all deaths worldwide originating from NCDs in 2016 [2], and inadequate habitual exercise in daily life leading to various NCDs, such as hypertension [3] and type 2 diabetes [4]. Studies have reported about health benefits of exercise not only for NCDs but also for 40 other items, namely mental health, accelerated biological aging, and premature death [5]. Furthermore, exercise has been reported to reduce the risk of infection and mortality from community-acquired infections and enhance immunity [6]. Thus, it can be considered as one of the positive habits that should be acquired to prevent further health damage associated with the new lifestyle of individuals owing to the COVID-19 pandemic and thus, live a healthy life.

The transtheoretical model (TTM) shows the process of behavior change, and in the case of habitual exercise, it is a flow that comprises the awareness of exercising, taking an initiative, continuing it, and making it a habit [7]. The TTM has been used in many studies as a useful indicator of behavioral change [8] and is often used in health checkups in Japan [9]. Furthermore, the percentage of those who have exercise habits is not high, and the percentage of Japanese men and women who self-reported that they have been exercising for a minimum of 30 min at least twice a week for more than a year is only 33.4% and 25.1%, respectively [10]. Contrastingly, a certain number of overestimations or underestimations have been reported regarding the perception of exercise implementation [11,12].

It has been suggested that steps of quarantine, aimed at preventing the further spread of COVID-19 infection has significantly lowered the risk of infection [13] but that it has also reduced habitual exercise [14]. Moreover, a systematic review of 14 countries reported a decrease in compliance with the WHO guidelines of physical activity from 80.9% to 62.5% compared to the pre-pandemic period [15]. Contrarily, positive associations between intention toward exercise behavior and physical activity [16] and the presence of highly active individuals [17] have been reported during governmental restrictions. Therefore, using TTM as an indicator, the implementation of awareness-raising activities for habitual exercise during the COVID-19 pandemic may be effective in increasing exercise frequency and thereby decreasing NCDs.

A survey using the Japanese International Physical Activity Questionnaire (IPAQ) reported a decrease in the duration of exercising after the declaration of a state of emergency for COVID-19 [18]. Regarding exercise, studies found that those who were aware of the benefits of improving their lifestyle may be able to convert the awareness into action even during the pandemic; thus, it is important to perceive the change in exercise behavior accompanying actual action. In behavior change, it is necessary to grasp the stage of TTM and provide guidance according to the stage [19]. For stage-based preintervention screening, it is important to examine the association between the perception of behavior change and habitual exercise. However, these associations are not well-understood, as are the characteristics of emergency declaration areas that specifically require self-quarantine. In this study, we examined the association between habitual exercise with the perception of behavior change using TTM and the state of emergency of the COVID-19 crisis.

## 2. Materials and Methods

### 2.1. Design

This study used the data obtained from the Kagoshima University Online Health Laboratory project (KU-OHL), a project for conducting an online questionnaire survey for adults living in Kagoshima Prefecture, analyzing the data, and providing health support. As a preliminary survey, we conducted a questionnaire survey of Internet users over the age of 20 living in Japan and received responses from 1600 people who applied randomly from 26 to 27 February 2021. In this online preliminary survey, we called on the registered users of Yahoo! Crowdsourcing, a crowdsourcing service started in 2013 by Yahoo Japan Corporation, Inc. (Tokyo, Japan), to participate in the research.

### 2.2. Sample

Of the 1600 participants who responded to the questionnaire survey (dataset 20210322), the study excluded those who exercised for more than 960 min/day (*N* = 16) and those who gave wrong answers (*N* = 85). Thus, 1499 participants (849 men and 650 women) were included in the final analysis.

### 2.3. Measures

The questionnaire survey collected information on age, sex, years of education, 1-year weight change, weight, height, food frequency, smoking habits, drinking habits, whether living alone or not, region (prefecture), household instability, medical history, prescribed medication, occupation, and perception of behavior change. The body mass index (BMI) (kg/m^2^) of the participants was calculated using their height and weight. As of 26–27 February 2021, the prefectures that were in a state of emergency due to the COVID-19 outbreak were Tokyo, Kanagawa, Saitama, Chiba, Osaka, Hyogo, Kyoto, Aichi, Gifu, and Fukuoka.

Standard questionnaire question items often used in Japanese health checkups [9] include “Are you going to start, or have you started lifestyle modifications?” (e.g., increase physical activity, improve dietary habits). “I do not mean to start” in the answer denotes TTM’s precontemplation, “I am going to start in the future (e.g., within 6 months)” denotes contemplation. However, “I am going to start soon (e.g., in a month) or I have just started some of them” denotes preparation, “I have already started (<6 months ago)” indicates action, and “I have already started (≥6 months ago)” indicates maintenance. A systematic review reported an association between TTM and exercise [20], and the validity of the TTM and exercise has been reported in a previous study targeting young women in Japan [21]. In this study, precontemplation and contemplation were defined as the absence of the perception of behavior change whereas preparation, action, and maintenance were defined as having a perception of behavior change for this question item, which exemplifies exercise. Habitual exercise was estimated based on the short version of the International Physical Activity Questionnaire (IPAQ). The amount of physical activity was categorized into the following three categories: walking, moderate physical activity, and vigorous physical activity. For each category, time exceeding 180 min/day was truncated to 180 min. The metabolic equivalent (MET) value was calculated by assigning the exercise intensity as 3.3 for “physical activity related to walking”, 4.0 for “physical activity related to moderate physical activity”, and 8.0 for “physical activity related to strong physical activity” and multiplying the frequency by time [22]. For habitual exercise, less than 600 MET min/week, which is less than the standard set by the WHO for physical activity, was defined as no habitual exercise while 600 MET min/week or more, which is above the standard, was considered as habitual exercise [23]. This IPAQ has been used for exercise evaluation in previous studies worldwide even before the outbreak of COVID-19, including Japan [24], and its validity and reliability has been confirmed in a previous study targeting Japanese participants [25].

Food frequency was evaluated using the food frequency score (FFS) [26,27]. The FFS represents the food frequency of the dietary variety score (DVS), which comprises food groups that account for about 80% of the staple food [28,29], side dishes, and soups that Japanese people usually eat. Prior study has reported the validity of DVS [30]. Furthermore, FFS is evaluated based on the weekly food intake frequency of 10 food groups (meat, seafood, eggs, soybean and soybean products, milk, green and yellow vegetables, seaweed, potatoes, fruits, and oil-based dishes). Eating almost every day, eating once every two days, eating once or twice a week, and hardly ever eating were given 3 points, 2 points, 1 point, and 0 points, respectively, and the total score was calculated based on the answers chosen by the participants. Furthermore, we determined the tertiles of the total score. Household instability was categorized into two groups based on the answers to the question “How do you feel about your current financial situation?”. No household instability was defined as responding with “I have an extremely comfortable family budget, and I live without worrying at all” or “I do not have a lot of financial resources, but I live without worrying” while “I do not have a budget, and I am a little worried” and “I am quite worried about my financial difficulties” were selected as indicators of household instability.

### 2.4. Analytic Strategy

In the comparison of the perception of behavior change, the χ^2^ test was used for categorical variables and the Mann–Whitney U test for continuous variables. Multivariate logistic regression analysis was utilized to determine the association between the perception of behavior change and habitual exercise, and the association of “OR > 1” was expressed as positive and “0 < OR < 1” as negative in the text. Sex, age (20–39, 40–64, ≥65 years), 1-year weight change (≤−3, nearly unchanged, ≥+3 kg), FFS (0–11, 12–16, ≥17 points), current smoking (No, Yes), state of emergency (No, Yes), household instability (No, Yes), medical history (No, Yes), and medication (No, Yes) were used as adjustment factors. Moreover, using the same analysis method, we performed an analysis excluding precontemplation and an analysis stratified by the state of emergency. All the statistical analyses were performed using Stata software version 15 (Stata Corp., College Station, TX, USA) with a statistical significance level of 5%.

## 3. Results

### 3.1. Characteristics of the Study Participants

The distribution of the participants was high in contemplation and in currently smoking, household instability, and self-rated health (Table 1). Precontemplation had a high proportion of those with a 1-year weight change nearly unchanged (kg), BMI (kg/m^2^) < 18.5, and medical history while preparation had a high proportion of those with years of education (years) ≥ 16, current drinking status, living alone or not, state of emergency, and occupation. The action was higher in women whereas maintenance was higher in FFS 0–11 points, medication, and median habitual exercise. In the perception of behavior change (No) group, age (years) 20–39, 1-year weight change (kg) nearly unchanged, current smoking, household instability, and medical history were significantly higher (Table 2). Meanwhile, in the perception of behavior change (Yes) group, the FFS of 0–11 points, the percentage of medication, and the median value of habitual exercise were significantly higher.

### 3.2. Association between the Perception of Behavior Change and Habitual Exercise

With habitual exercise as the dependent variable, the odds ratio (OR) of the independent variable perception of behavior change was positively associated with crude OR (2.62, 2.06–3.32) and adjusted OR (2.41, 1.89–3.08) (Table 3). The independent variables in the same analysis were crude OR (0.71, 0.57–0.88) and adjusted OR (0.63, 0.49–0.80) for women, crude OR (0.71, 0.54–0.92) and adjusted OR (0.76, 0.57–1.00) for FFS ≥ 17 points, crude OR (1.73, 1.11–2.71) and adjusted OR (1.64, 1.03–2.62) for 1-year weight change (kg) ≥ +3, and crude OR (1.39, 1.07–1.80) and adjusted OR (1.33, 1.02–1.75) for FFS 0–11 points. However, medical history was significantly negatively associated with habitual exercise but not in adjusted OR.

In the same analysis excluding precontemplation, OR for the perception of behavior change was crude OR (2.50, 1.91–3.27) and adjusted OR (2.30, 1.74–3.04), and 1-year weight change (kg) ≥ +3 was crude OR (1.94, 1.10–3.41) and adjusted OR (1.81, 1.01–3.25) (Table 4). In women, crude OR (0.68, 0.52–0.89) and adjusted OR (0.65, 0.49–0.87) whereas FFS ≥ 17 points showed crude OR (0.64, 0.46–0.89) and adjusted OR (0.69, 0.49–0.97).

In the state of non-emergency during the COVID-19 pandemic, the perception of behavior change for habitual exercise was crude OR (2.29, 1.57–3.34) and adjusted OR (2.01, 1.34–3.01); however, crude OR (2.86, 2.10–3.88) and adjusted OR (2.69, 1.97–3.69) were observed for the state of emergency (Table 5). In women, both crude OR and adjusted OR were significantly negatively associated with habitual exercise except for crude OR in the state of non-emergency. Only in crude OR, FFS ≥ 17 points, and medical history in state of emergency were negatively associated with habitual exercise.

## 4. Discussion

In this study, we observed the association between habitual exercise and the perception of behavior change in internet users among adults living in Japan during the COVID-19 pandemic and found a significant positive association. The results indicate that the perception of behavior change contributes to habitual exercise during the pandemic.

According to a survey conducted in Japan before the outbreak of COVID-19, the percentage of people who consciously exercised was (1) 51.8% of men and 53.1% of women in 1996, (2) 54.2% of men and 55.5% of women in 2003, and (3) 58.7% men and 60.5% women in 2008, with significant increases for both men and women [31]. The perception of behavior change in this study showed a combined low value of 39.6% for both men and women. Quantitative and qualitative studies have reported an increase in the awareness of the benefits of indoor exercise during the COVID-19 pandemic [32,33], however, some people are less likely to shift their awareness of exercise to practice.

Another data reported by the Health Japan 21 Evaluation Work Team on the proportion of habitual exercisers who exercised for 30 min or more twice a week for at least one year showed no significant increase, with 28.6% men and 24.6% women in 1997, 30.9% men and 25.8% women in 2004, and 32.2% men and 27.0% women in 2009 [31]. Among the participants of this study, the maintenance of TTM, which indicates exercise habituation, was 14.5%. As an increase in sedentary behavior [34] and a decrease in the amount of exercise have been reported during the COVID-19 crisis [35], it is possible that the proportion of those who had a habit of exercising was low in this study as well.

While using TTM for exercise so far [20], the participant’s misperception regarding exercise awareness and behavior has also been reported [36,37]. Furthermore, these results differed by BMI and sex, with exercise overestimation observed in those with low BMI and those of normal weight [38,39]. There is also a previous study reporting that women do less physical activity [40]. The findings of this study confirmed a positive association between habitual exercise and the perception of behavior change. By excluding TTM precontemplation, a sub-analysis that adjusted for the influence of so-called “[those who have a habit of exercising] who are considered to be good and have no intention of changing their exercise habits” also showed a positive association between habitual exercise and the perception of behavior change. This same association was recognized regardless of the state of emergency declared due to the pandemic. These results are reasonable findings in TTM. Regarding exercise during the pandemic, some studies in Europe and the United States have reported an increase in exercise [41,42]. Furthermore, a positive association between health consciousness and home-based exercise has been reported in Asia [43]. Although previous research is limited in this way, there may be an increase in exercises in situations such as the COVID-19 pandemic, and there may be an involvement in the habitual exercise of perception of behavior change, similar to the results of this study. Furthermore, women were associated with less habitual exercise. There is no biological plausibility for these associations, thereby suggesting possible social implications. Previous studies have reported that women spent more time on household chores and childcare than men during the COVID-19 pandemic [44] and were involved in providing care for their loved ones [45]. Similar reports have also been made by the Cabinet Office of Japan [46]. Except for the stratified analysis by the state of emergency, 1-year weight change ≥+3 kg was positively associated, and FFS ≥ 17 points were negatively associated with habitual exercise. Previous studies on adults have reported weight gain during COVID-19 [47,48], whereas weight loss in general has been associated with diet or exercise [49]. It is hard to believe that those who gained 3 kg or more in one year gained weight because they were doing habitual exercise. Habitual exercise due to weight gain suggests a reverse causal effect. Regarding FFS, food intake frequency with 17 points or more was negatively associated with habitual exercise. There are limited reports on the association between food intake frequency and habitual exercise. A study of people with type-2 diabetes during the lockdown reported more frequent vegetable intake and less exercise [50]. Furthermore, previous research that conducted a random telephone survey of adults affected by COVID-19 reported a high intake of vegetables and fruits and a decrease in exercise [51]. The FFS used in this study included the frequency of vegetable and fruit intake, and the results obtained were similar to those reported on diet and exercise. Furthermore, these associations were not observed regardless of the state of emergency. Previous studies on these associations are also limited; thus, the validity of the findings of this study is unclear; however, it is suggested that the effect of the decrease in the number of participants is due to stratification.

This study has some limitations. One limitation of this study is that the estimates of habitual exercise are self-reported, and the estimates using the short version of the IPAQ could be slightly overestimated [22]. Perception of behavior change may have been misclassified (e.g., recall bias), and the results may have been diluted. Therefore, we examined whether the interpretation of the findings of this study is consistent with reference to the results of previous studies and biological validity. Furthermore, in this study, we conducted a questionnaire survey that collected a wide range of lifestyle information and adjusted for confounding factors; however, there is a possibility that the existence of unknown confounding factors could not be adjusted. Thus, it is necessary to especially consider the influence of social implications.

## 5. Conclusions

This study cross-sectionally examined the association between the perception of behavior change and habitual exercise among Internet users aged 20–86 years living in Japan during the COVID-19 pandemic. Perception of behavior change was suggested to be positively associated with habitual exercise regardless of the COVID-19 state of emergency. As for related factors, an association between women and less habitual exercise was found. The findings of this study will contribute to the screening of habitual exercise based on the perception of behavior change in the COVID-19 pandemic and the implementation of educational activities for acquiring habits of regular exercise. In the future, it will be thus necessary to verify the association between the perception of behavior change and habitual exercise during the COVID-19 pandemic or other pandemics using physical activity measurements.

## Figures and Tables

**Table 1 ijerph-20-00356-t001:** Characteristics of study participants according to transtheoretical model.

	Transtheoretical Model									
	Precontemplation (*n* = 427)		Contemplation (*n* = 478)		Preparation (*n* = 213)		Action (*n* = 163)		Maintenance (*n* = 218)	
	*N* (%)									
Age (in years)										
20–39	199	(46.6)	213	(44.6)	108	(50.7)	68	(41.7)	70	(32.1)
40–64	168	(39.3)	228	(47.7)	90	(42.3)	73	(44.8)	92	(42.2)
≥65	60	(14.1)	37	(7.7)	15	(7.0)	22	(13.5)	56	(25.7)
Women	166	(38.9)	226	(47.3)	90	(42.3)	85	(52.2)	83	(38.1)
Years of education (years)										
≤9	3	(0.7)	4	(0.8)	2	(0.9)	3	(1.8)	5	(2.3)
10–15	203	(47.5)	216	(45.2)	85	(39.9)	71	(43.6)	89	(40.8)
≥16	221	(51.8)	258	(54.0)	126	(59.2)	89	(54.6)	124	(56.9)
1-year weight change (kg)										
≤−3	50	(11.7)	98	(20.5)	59	(27.7)	39	(23.9)	30	(13.8)
Nearly unchanged	348	(81.5)	352	(73.6)	142	(66.7)	105	(64.4)	160	(73.4)
≥+3	29	(6.8)	28	(5.9)	12	(5.6)	19	(11.7)	28	(12.8)
BMI (kg/m^2^)										
<18.5	77	(37.2)	59	(28.5)	21	(10.1)	15	(7.3)	35	(16.9)
≥18.5 to <25.0	289	(28.3)	323	(31.6)	143	(14.0)	111	(10.9)	155	(15.2)
≥25.0	61	(22.5)	96	(35.4)	49	(18.1)	37	(13.7)	28	(10.3)
FFS (points)										
0–11	144	(33.7)	158	(33.1)	93	(43.7)	68	(41.7)	111	(50.9)
12–16	143	(33.5)	167	(34.9)	72	(33.8)	58	(35.6)	67	(30.7)
≥17	140	(32.8)	153	(32.0)	48	(22.5)	37	(22.7)	40	(18.4)
Current smoking status	100	(23.4)	112	(23.4)	43	(20.2)	34	(20.9)	32	(14.7)
Current drinking status	280	(65.6)	343	(71.8)	160	(75.1)	120	(73.6)	146	(67.0)
Living alone	85	(19.9)	87	(18.2)	44	(20.7)	25	(15.3)	43	(19.7)
State of emergency	265	(62.1)	305	(63.8)	142	(66.7)	95	(58.3)	137	(62.8)
Household instability	159	(37.2)	215	(45.0)	79	(37.1)	60	(36.8)	75	(34.4)
Medical history	215	(50.4)	210	(43.9)	89	(41.8)	58	(35.6)	67	(30.7)
Medication	112	(26.2)	143	(29.9)	71	(33.3)	69	(42.3)	106	(48.6)
Occupation	367	(86.0)	415	(86.8)	195	(91.6)	146	(89.6)	168	(77.1)
Self-rated health	360	(28.7)	388	(30.9)	180	(14.3)	142	(11.3)	186	(14.8)
	Median (range)									
Habitual exercise (MET·min/week)	960	0–17,496	996	0–13,112	1386	0–16,853	1748	0–13,083	2360	0–19,278

BMI: Body mass index; FFS: Food frequency score; MET: Metabolic equivalent.

**Table 2 ijerph-20-00356-t002:** Comparison of participants’ characteristics according to the perception of behavior change.

	Perception of Behavior Change				
	No (*n* = 905)		Yes (*n* = 594)		*p*
	*N* (%)				
Age (in years)					
20–39	412	(45.5)	246	(41.4)	0.015 ^†^
40–64	396	(43.8)	255	(42.9)	
≥65	97	(10.7)	93	(15.7)	
Women	392	(43.3)	258	(43.4)	0.964 ^†^
Years of education (years)					
≤9	7	(0.8)	10	(1.7)	0.055 ^†^
10–15	419	(46.3)	245	(41.3)	
≥16	479	(52.9)	339	(57.1)	
1-year weight change (kg)					
≤−3	57	(6.3)	59	(9.9)	0.001 ^†^
Nearly unchanged	700	(77.4)	407	(68.5)	
≥+3	148	(16.4)	128	(21.6)	
BMI (kg/m^2^)					
<18.5	136	(15.0)	71	(12.0)	0.201^†^
≥18.5 to <25.0	612	(67.6)	409	(68.9)	
≥25.0	157	(17.4)	114	(19.2)	
FFS (points)					
0–11	302	(33.4)	272	(45.8)	<0.001 ^†^
12–16	310	(34.3)	197	(33.2)	
≥17	293	(32.4)	125	(21.0)	
Current smoking status	212	(23.4)	109	(18.4)	0.019 ^†^
Current drinking status	623	(68.8)	426	(71.7)	0.235 ^†^
Living alone	172	(19.0)	112	(18.9)	0.942 ^†^
State of emergency	570	(63.0)	374	(63.0)	0.994 ^†^
Household instability	374	(41.3)	214	(36.0)	0.040 ^†^
Medical history	425	(47.0)	214	(36.0)	<0.001 ^†^
Medication	255	(28.2)	246	(41.4)	<0.001 ^†^
Occupation	782	(86.4)	509	(85.7)	0.694 ^†^
Self-rated health	748	(82.7)	508	(85.5)	0.140 ^†^
	Median (range)				
Habitual exercise (MET·min/week)	990	0–17,496	1799	0–19,278	<0.001 ^††^

BMI: Body mass index; FFS: Food frequency score; MET: Metabolic equivalent. ^†^ χ^2^ test. ^††^ Mann–Whitney U test.

**Table 3 ijerph-20-00356-t003:** Association between the perception of behavior change and habitual exercise.

	Habitual Exercise (*n* = 1499)					
	Crude OR	95%CI		Adjusted OR	95%CI	
Perception	2.62	2.06	3.32	2.41	1.89	3.08
Age (in years)						
20–39	1.06	0.84	1.33	1.18	0.92	1.52
40–64	1.00			1.00		
≥65	1.22	0.86	1.73	0.94	0.64	1.37
Women	0.71	0.57	0.88	0.63	0.49	0.80
1-year weight change (kg)						
≤−3	1.10	0.83	1.46	1.00	0.74	1.34
Nearly unchanged	1.00			1.00		
≥+3	1.73	1.11	2.71	1.64	1.03	2.62
FFS (points)						
0–11	1.39	1.07	1.80	1.33	1.02	1.75
12–16	1.00			1.00		
≥17	0.71	0.54	0.92	0.76	0.57	1.00
Current smoking status	0.89	0.69	1.15	0.93	0.70	1.23
State of emergency	1.14	0.91	1.42	1.16	0.92	1.46
Household instability	1.18	0.95	1.47	1.09	0.86	1.37
Medical history	0.76	0.61	0.95	0.83	0.65	1.06
Medication	1.23	0.98	1.55	1.03	0.80	1.34

FFS: Food frequency score; OR: Odds ratio; CI: Confidence Interval.

**Table 4 ijerph-20-00356-t004:** Association between the perception of behavior change and habitual exercise without precontemplation.

	Habitual Exercise (*n* = 1072)					
	Crude OR	95%CI		Adjusted OR	95%CI	
Perception	2.50	1.91	3.27	2.30	1.74	3.04
Age (in years)						
20–39	1.05	0.79	1.39	1.17	0.86	1.58
40–64	1.00			1.00		
≥65	1.13	0.74	1.74	0.77	0.48	1.24
Women	0.68	0.52	0.89	0.65	0.49	0.87
1-year weight change (kg)						
≤−3	1.00	0.72	1.37	0.91	0.65	1.28
Nearly unchanged	1.00			1.00		
≥+3	1.94	1.10	3.41	1.81	1.01	3.25
FFS (points)						
0–11	1.29	0.94	1.77	1.24	0.90	1.73
12–16	1.00			1.00		
≥17	0.64	0.46	0.89	0.69	0.49	0.97
Current smoking status	1.00	0.73	1.39	1.05	0.74	1.49
State of emergency	1.26	0.96	1.65	1.32	1.00	1.75
Household instability	1.22	0.93	1.59	1.11	0.84	1.47
Medical history	0.78	0.60	1.02	0.83	0.62	1.12
Medication	1.21	0.92	1.59	1.06	0.77	1.44

FFS: Food frequency score.

**Table 5 ijerph-20-00356-t005:** Association between the perception of behavior change and habitual exercise according to state of emergency.

	Habitual Exercise											
	State of Non-Emergency (*n* = 555)						State of Emergency (*n* = 944)					
	Crude OR	95%CI		Adjusted OR	95%CI		Crude OR	95%CI		Adjusted OR	95%CI	
Perception	2.29	1.57	3.34	2.01	1.34	3.01	2.86	2.10	3.88	2.69	1.97	3.69
Age (in years)												
20–39	1.15	0.79	1.66	1.22	0.82	1.83	1.01	0.75	1.35	1.16	0.84	1.60
40–64	1.00			1.00			1.00			1.00		
≥65	1.44	0.80	2.62	1.29	0.68	2.42	1.10	0.71	1.70	0.78	0.48	1.25
Women	0.74	0.52	1.06	0.65	0.44	0.96	0.68	0.52	0.90	0.62	0.45	0.84
1-year weight change (kg)												
≤−3	1.55	0.96	2.50	1.33	0.80	2.20	0.91	0.64	1.29	0.88	0.61	1.26
Nearly unchanged	1.00			1.00			1.00			1.00		
≥+3	1.65	0.83	3.30	1.48	0.72	3.05	1.80	1.00	3.26	1.82	0.98	3.37
FFS (points)												
0–11	1.40	0.92	2.13	1.34	0.87	2.08	1.38	0.99	1.92	1.32	0.94	1.87
12–16	1.00			1.00			1.00			1.00		
≥17	0.80	0.52	1.24	0.84	0.54	1.33	0.65	0.47	0.92	0.71	0.50	1.01
Current smoking status	0.86	0.57	1.32	0.96	0.60	1.52	0.90	0.65	1.26	0.90	0.63	1.28
Household instability	1.23	0.86	1.75	1.12	0.77	1.62	1.14	0.86	1.51	1.08	0.80	1.45
Medical history	0.82	0.58	1.17	0.93	0.62	1.40	0.72	0.55	0.95	0.78	0.57	1.07
Medication	1.24	0.85	1.81	1.03	0.67	1.59	1.22	0.91	1.64	1.06	0.76	1.47

FFS: Food frequency score.

## Data Availability

The data used in this study does not have permission to link and share for ethical reasons.

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
