# Peer review of "Association between the Perception of Behavior Change and Habitual Exercise during COVID-19: A Cross-Sectional Online Survey in Japan"

_ijerph, 2022, doi:10.3390/ijerph20010356_

Round 1
Reviewer 1 Report
Comments
The work has social value insofar as it allows to increase the knowledge and advantages of the habitual exercise. In this case, it is about collecting information on the perception of behavior change during COVID-19. It is proposed to examine the association between habitual exercise with the perception of behavior change using TTM and the state of emergency of the COVID-19 crisis.
When reading the introduction, a first question. There are other theoretical models of behavior change, why use TTM and not others. In addition, another instrument was used to assess habitual exercise is self-reported (IPAQ)
In this regard, and having read the limitations of the work reported by the researchers, the following questions arise: If both instruments were validated? If memory bias and other possible biases that could influence the results were considered?
Regarding the measurement of perception with a TTM theory approach, it requires that there be an intervention in this regard beforehand, after a baseline, and then measure results at 3, 6, and 9 months. Changes are supposed to be experienced within the first 6 months. In this case it is not clear how it was evaluated and if any type of intervention was considered, especially directed at the purpose of the study and not only as an individual initiative. This to be able to evaluate.
In the regression tables, ORs were obtained whose results could be interpreted as protective factors, but there are no comments on this.
Among the results it is suggested that women face difficulties to carry out habitual exercise. Regarding this, some elements that are cultural of the country and that affect not only the issue of exercise but also another sphere of society could be considered.
Author Response
Dear reviewer
We received comments and revised the manuscript. This manuscript received a native check again. We will resubmit the manuscript. Please see the attachment.
Also, comments are listed below.
Best regards,
Nishimoto
IJERPH Comments
Reviewer 1
Thank you very much for the opportunity to review the manuscript. It has been very interesting. I would like to make a few comments with the intention of providing the best information to the readers:
Response
Thank you for your comment. We appreciate your helpful and insightful comments on our manuscript. Please see our responses to your comments.
The work has social value insofar as it allows to increase the knowledge and advantages of the habitual exercise. In this case, it is about collecting information on the perception of behavior change during COVID-19. It is proposed to examine the association between habitual exercise with the perception of behavior change using TTM and the state of emergency of the COVID-19 crisis.
When reading the introduction, a first question. There are other theoretical models of behavior change, why use TTM and not others. In addition, another instrument was used to assess habitual exercise is self-reported (IPAQ)
Response 1
Thank you for your comment. We used the term TTM because it has been used in many studies as a useful indicator of behavioral change and is often used in health checkups in Japan.
The introduction part has been modified as follows.
Location in the text
Line 45-47:
The TTM has been used in many studies as a useful indicator of behavioral change [8] and is often used in health checkups in Japan [9].
Response 2
Thank you for your comment. We used IPAQ because it has been used for exercise evaluation in previous studies around the world, including Japanese. I have now added the following in the Materials and Methods part.
Location in the text
Line 123-25:
This IPAQ has been used for exercise evaluation in previous studies worldwide even before the outbreak of COVID-19, including Japan [24], and its validity and reliability has been confirmed in previous study targeting Japanese participants [25].
In this regard, and having read the limitations of the work reported by the researchers, the following questions arise: If both instruments were validated? If memory bias and other possible biases that could influence the results were considered?
Response 1
Thank you for your comment. The association between TTM and exercise has been reported in a systematic review, and the validity of TTM and exercise has been reported in a previous study targeting young women in Japan. Therefore, we added the following in the Materials and Methods part.
Location in the text
Line 105:
A systematic review reported an association of TTM and exercise [20], and the validity of the TTM and exercise has been reported in a previous study targeting young women in Japan [21].
Response 2
Thank you for your comment. The validity and reliability of IPAQ translated into Japanese has been confirmed in a previous study targeting Japanese people, and we added it as follows.
Location in the text
Line 123:
…and its validity and reliability has been confirmed in previous study targeting Japanese participants [25].
Response 3
Thank you for your comment. Regarding bias, we have revised the description of the limitation part based on your comment.
Location in the text
Line 296:
Perception of behavior change may have been misclassified (e.g., recall bias), and the results may have been diluted. …Furthermore, in this study, we conducted a questionnaire survey that collected a wide range of lifestyle information and adjusted for confounding factors, but there is a possibility that the existence of unknown confounding factors could not be adjusted.
Regarding the measurement of perception with a TTM theory approach, it requires that there be an intervention in this regard beforehand, after a baseline, and then measure results at 3, 6, and 9 months. Changes are supposed to be experienced within the first 6 months. In this case it is not clear how it was evaluated and if any type of intervention was considered, especially directed at the purpose of the study and not only as an individual initiative. This to be able to evaluate.
Response
Thank you for your comment. We added to the introduction part that this study examines the association of perception of behavior change and habitual exercise in preintervention screening.
Location in the text
Line 67:
In behavior change, it is necessary to grasp the stage of TTM and provide guidance according to the stage [19]. For stage-based preintervention screening, it is important to examine the association between perception of behavior change and habitual exercise.
In the regression tables, ORs were obtained whose results could be interpreted as protective factors, but there are no comments on this.
Response
Thank you for your comment. We noticed that the expression of positive and negative based on OR was insufficient, so we expressed the association of "OR >1" as positive and the association of "0< OR <1" as negative in the Materials and Methods part.
Location in the text
Line 145-147:
Multivariate logistic regression analysis was utilized to determine the association be-tween the perception of behavior change and habitual exercise, and the association of "OR >1" was expressed as positive and "0< OR <1" as negative in the text.
Among the results it is suggested that women face difficulties to carry out habitual exercise. Regarding this, some elements that are cultural of the country and that affect not only the issue of exercise but also another sphere of society could be considered.
Response
Thank you for your comment. In addition to the discussion part "There is no biological plausibility for these associations, thereby suggesting possible social implications," we added the sentence " Thus, it is necessary to especially consider the influence of social implication." in the limitation part.
Location in the text
Line 300:
Thus, it is necessary to especially consider the influence of social implication.

Reviewer 2 Report
Dear authors, congratulations on your excellent results and publication. The article made a very positive impression on me. At the same time, I would like to clarify only a few issues
1. More information about the IPAQ should obviously be provided, namely whether a translation into Japanese was made within the scope of this study or earlier
2. Calculating IPAQ results involves screening out, for example, results that are too high (for example, the duration is greater than the number of hours in a day). Was such an analysis of the obtained data done within the scope of this study?
Kind regards,
Author Response
Dear reviewer 2
We received comments and revised the manuscript. This manuscript received a native check again. We will resubmit the manuscript. Please see the attachment.
Also, comments are listed below.
Best regards,
Nishimoto
IJERPH Comments
Reviewer 2
Thank you very much for the opportunity to review the manuscript. It has been very interesting. I would like to make a few comments with the intention of providing the best information to the readers:
Response
Thank you for your comment. We appreciate your helpful and insightful comments. Please see our responses to your comments.
- More information about the IPAQ should obviously be provided, namely whether a translation into Japanese was made within the scope of this study or earlier
Response
Thank you for your comment. Thank you for your valuable comment. In response to your comment, we have added the following sentence to the Materials and Methods part. In addition, I added a previous study using IPAQ to the Introduction part.
Location in the text
Line 121 (Materials and Methods part):
This IPAQ has been used for exercise evaluation in previous studies around the world even before the outbreak of COVID-19, including Japan [24], and its validity and reliability has been confirmed in previous study targeting Japanese participants [25].
Line 62 (Introduction part):
A survey using the Japanese International Physical Activity Questionnaire (IPAQ) reported a decrease in exercise time after the declaration of a state of emergency for COVID-19 [18].
- Calculating IPAQ results involves screening out, for example, results that are too high (for example, the duration is greater than the number of hours in a day). Was such an analysis of the obtained data done within the scope of this study?
Response
Thank you for your comment. We have corrected the article. In this study, we excluded 16 patients with over 960 minutes/day as too high results on IPAQ, and the analysis was based on reference 22. Similarly, for each of walking, moderate physical activity, and vigorous physical activity, time exceeding 180 minutes/day was rounded down to 180 minutes.
Location in the text
Line 86:
Of the 1600 participants who responded to the questionnaire survey (dataset 20210322), the study excluded those who exercised for more than 960 minutes/day (N=16) and …
Line 114:
For each category, time exceeding 180 minutes/day was truncated to 180 minutes.

Reviewer 3 Report
The topic is interesting although largely investigated int the extensive COVID-19 literature. The design has limited impact. Conclusions are fairly understandable-
Please provide a point-by-point response to the following issues.
1) The introduction should provide background and information relevant to the study. Comparison with similar studies in literature should be done. Limited comparable European studies were reported. Please see Chirico et al. (10.3389/fpsyg.2020.02100) and Tornaghi et al. (10.23736/S0022-4707.20.11600-1) as regards Italy, for instance.
2) The title should reflect the design of the study.
Material and Methods
3. The general weakeness (which should be acknowledged) relates to the fact that all responses are subjective, and all subjects completed the survey at one timepoint so the findings are subject to recall bias.
4/ The manuscript should provide more information about the measures and the measured constructs in more general terms. What is the theoretical framework behind the developed questionnaire (e.g. Is it reliable? How can we be sure about what is realiabily mesured by the questionnaire? is there any unitary construct behind it ?).
5) Results
Results are peculiar of the population studied and might be appreciable in the context of other similar European studies.
Author Response
Dear reviewer 3
We received comments and revised the manuscript. This manuscript received a native check again. We will resubmit the manuscript. Please see the attachment.
Also, comments are listed below.
Best regards,
Nishimoto
IJERPH Comments
Reviewer 3
Thank you very much for the opportunity to review the manuscript. It has been very interesting. I would like to make a few comments with the intention of providing the best information to the readers:
Response
Thank you for your comment. We appreciate your helpful and insightful comments. Please see our responses to your comments.
1) The introduction should provide background and information relevant to the study. Comparison with similar studies in literature should be done. Limited comparable European studies were reported. Please see Chirico et al. (10.3389/fpsyg.2020.02100) and Tornaghi et al. (10.23736/S0022-4707.20.11600-1) as regards Italy, for instance.
Response
Thank you for the precious opinion. We provided the background in the introduction based on the papers you suggested.
Location in the text
Line 57:
On the other hand, positive associations between intention toward the behavior and physical activity [16] and presence of highly active individuals [17] have been reported during governmental restrictions.
2) The title should reflect the design of the study.
Response
Thank you for your comment. The title has been changed to reflect the research design.
Location in the text
Line 2:
Association between the perception of behavior change and habitual exercise during COVID-19: A cross-sectional online survey in Japan
- The general weakeness (which should be acknowledged) relates to the fact that all responses are subjective, and all subjects completed the survey at one timepoint so the findings are subject to recall bias.
Response
Thank you for your comment. Regarding bias, we have revised the description of the limitation part based on your comment.
Location in the text
Line 297:
Perception of behavior change may have been misclassified (e.g., recall bias), and the results may have been diluted. …
4/ The manuscript should provide more information about the measures and the measured constructs in more general terms. What is the theoretical framework behind the developed questionnaire (e.g. Is it reliable? How can we be sure about what is realiabily mesured by the questionnaire? is there any unitary construct behind it ?).
Response
Thank you for your comment. Regarding the developed questionnaire, we added background, the reliability and validity of exercise to Japanese people to the text explaining IPAQ. Subsequently, the TTM presented the process of behavioral change in the introduction part and was limited to describing only five structures in the Materials and Methods part. Therefore, we added details about background and the validity of exercise by TTM. In addition, we added that FFS shows the food frequency of a valid scale (Dietary Variety Score: DVS).
Location in the text
Line 121:
This IPAQ has been used for exercise evaluation in previous studies around the world even before the outbreak of COVID-19, including Japan [24], and its validity and reliability has been confirmed in previous study targeting Japanese participants [25].
Line 105:
A systematic review reported an association of TTM and exercise [20], and the validity of the TTM and exercise has been reported in a previous study targeting young women in Japan [21].
Line 125:
FFS represent the food frequency of the dietary variety score (DVS), which consists of food groups that account for about 80% of the staple food [28,29], side dishes, and soups that Japanese people usually eat, and the validity of DVS has been reported [30]. In addition, …
5) Results
Results are peculiar of the population studied and might be appreciable in the context of other similar European studies.
Response 1
Thank you for your comment. In response to your comment, we referred to a study reporting an increase in exercises of the covid-19 pandemic in the West. In addition, we referred to a previous study that reported a positive association between health consciousness and home-based exercise in Asia (discussion part).
Location in the text
Line 266:
Regarding the exercise of the covid-19 pandemic, some studies in Europe and the United States have reported an increase in exercise [41,42], and a positive association between health consciousness and home-based exercise has been reported in Asia [43]. Although previous research is limited in this way, there may be an increase in exercises in situations such as the covid-19 pandemic, and there may be an involvement in the habitual exercise of perception of behavior change, similar to the results of this study.
Response 2
Thank you for your comment. We have deemed it more appropriate to use reference 43 in the above text and have deleted the old L257 text below.
Location in the text
Line 256:
Before revision) Among the men and women affected by COVID-19, more men exercise at home [previously 29: present 43], and there is also a previous study reporting that women do less physical activity [previously 30: present 40].
After revision) There is also a previous study reporting that women do less physical activity [previously 30: present 40].

Round 2
Reviewer 3 Report
I have read the revised manuscript and the response to reviewers. I appreciate that the authors provided arguments and improved the manuscript according to the reviewers’ comments. I am satisfied with the manuscript and authors’ responses. I have no further request. Thank you.